# Elevation Influences Belowground Biomass Proportion in Forests by Affecting Climatic Factors, Soil Nutrients and Key Leaf Traits

**DOI:** 10.3390/plants13050674

**Published:** 2024-02-28

**Authors:** Xing Zhang, Yun Wang, Jiangfeng Wang, Mengyao Yu, Ruizhi Zhang, Yila Mi, Jiali Xu, Ruifang Jiang, Jie Gao

**Affiliations:** 1Key Laboratory for the Conservation and Regulation Biology of Species in Special Environments, College of Life Science, Xinjiang Normal University, Urumqi 830054, China; zxyybh@163.com (X.Z.); www030517@outlook.com (Y.W.); wjf2088683747@163.com (J.W.); yao02292023@163.com (M.Y.); zrz13699361635@163.com (R.Z.); 18999541576@163.com (Y.M.); m18099534368@163.com (J.X.); 2Xinjiang Uyghur Autonomous Region Forestry Planning Institute, Urumqi 830046, China; ecoirf@163.com; 3Key Laboratory of Earth Surface Processes of Ministry of Education, College of Urban and Environmental Sciences, Peking University, Beijing 100863, China

**Keywords:** belowground biomass proportion, elevation, climate change, soil nutrient, key leaf traits

## Abstract

Forest biomass allocation is a direct manifestation of biological adaptation to environmental changes. Studying the distribution patterns of forest biomass along elevational gradients is ecologically significant for understanding the specific impacts of global change on plant resource allocation strategies. While aboveground biomass has been extensively studied, research on belowground biomass remains relatively limited. Furthermore, the patterns and driving factors of the belowground biomass proportion (BGBP) along elevational gradients are still unclear. In this study, we investigated the specific influences of climatic factors, soil nutrients, and key leaf traits on the elevational pattern of BGBP using data from 926 forests at 94 sites across China. In this study, BGBP data were calculated from the root biomass to the depth of 50 cm. Our findings indicate considerable variability in forest BGBP at a macro scale, showing a significant increasing trend along elevational gradients (*p* < 0.01). BGBP significantly decreases with increasing temperature and precipitation and increases with annual mean evapotranspiration (MAE) (*p* < 0.01). It decreases significantly with increasing soil phosphorus content and increases with soil pH (*p* < 0.01). Key leaf traits (leaf nitrogen (LN) and leaf phosphorus (LP)) are positively correlated with BGBP. Climatic factors (R^2^ = 0.46) have the strongest explanatory power for the variation in BGBP along elevations, while soil factors (R^2^ = 0.10) and key leaf traits (R^2^ = 0.08) also play significant roles. Elevation impacts BGBP directly and also indirectly through influencing such as climate conditions, soil nutrient availability, and key leaf traits, with direct effects being more pronounced than indirect effects. This study reveals the patterns and controlling factors of forests’ BGBP along elevational gradients, providing vital ecological insights into the impact of global change on plant resource allocation strategies and offering scientific guidance for ecosystem management and conservation.

## 1. Introduction

Biomass is a direct indicator of a plant’s carbon sequestration capability [1]. Previous studies have primarily focused on aboveground biomass [2,3,4], often overlooking the ecological contribution of belowground biomass. Environmental factors at different elevational gradients, such as temperature and precipitation, exhibit significant variations [5]. Studying the distribution patterns of forest biomass along elevational gradients is ecologically important for understanding the specific impacts of global change on plant resource allocation strategies [6]. The patterns of plant biomass variation along elevational gradients have been preliminarily studied [7,8]. He et al. (2017) found that the aboveground biomass of *Gentiana Hexagonia* in the eastern Tibetan Plateau first increases and then decreases with elevation [9]. Research by Zhang et al. observed that forest stem and root biomass significantly increase with elevation, while leaf biomass does not exhibit a clear elevational trend [10]. Some studies have also found that with increasing elevation, herbaceous plants reduce the proportion of aboveground biomass allocation, allocating more biomass to belowground parts [7,11]. However, due to the difficulty of obtaining belowground biomass data from forest trees in field surveys, the patterns and dominant factors of the belowground biomass proportion (BGBP) along elevational gradients on a macro scale remain unclear.

Along with elevational gradients, climatic factors, particularly temperature and precipitation, undergo significant changes. Extensive research has found that tree aboveground–belowground biomass allocation strategies are regulated by climatic factors [12,13]. As temperature decreases, plants’ BGBP significantly increases [14]. With a reduction in temperature, the efficiency of leaf photosynthesis decreases, leading to a reduction in energy and carbohydrate production and potentially less growth investment in aboveground parts [15]. Under cooler conditions, nutrient mobility and microbial activity in the soil are also typically reduced [16], prompting plants to develop larger root systems for effective nutrient absorption, thereby increasing BGBP. BGBP also significantly increases with decreasing precipitation. When rainfall diminishes, plants face water stress and expand their root systems for the more efficient absorption and utilization of limited water resources, exploring a larger soil volume to access water [17,18]. In arid conditions, plants reduce their aboveground growth to decrease transpiration and minimize water loss [19]. Besides thermal and moisture factors, daylight duration is another key climatic factor affecting BGBP. With increased daylight duration, the duration of plant photosynthesis increases, as does the demand for nutrients, especially elements like nitrogen and phosphorus [20]. To effectively absorb these nutrients, plants enhance root growth, increasing BGBP [21,22]. Additionally, longer daylight hours often coincide with higher temperatures, which may affect soil moisture evaporation and nutrient availability. Plants might adapt to these changes by increasing root biomass [23].

With changes in elevational gradients, soil nutrients also undergo significant alterations [24]. Soil, as the direct habitat for plants, has an important influence on BGBP due to the availability of soil nutrients. BGBP significantly decreases with an increase in soil nitrogen and phosphorus [25]. As soil nitrogen and phosphorus increase, these key nutrients become more easily accessible to plants, reducing the need for plants to expend energy on expanding their root systems to acquire these nutrients and, hence, significantly decreasing BGBP. Moreover, nitrogen and phosphorus are essential for photosynthesis, and their increased availability enhances photosynthetic efficiency, leading plants to allocate more resources to aboveground parts to support rapid growth [26,27]. Soil pH is another key soil factor influencing BGBP. For example, in alkaline environments, the availability of many nutrients is reduced, prompting plants to develop their root systems further to acquire more nutrients [28]. Additionally, alkaline soil affects the composition and activity of soil microbial communities, reducing the decomposition of organic matter. Plants adapt to alkaline conditions by increasing root biomass [29].

Elevation significantly affects key leaf traits in plants [30]. Key leaf traits directly influence photosynthesis and are important biological factors affecting plant biomass allocation strategies [31]. The specific leaf area (SLA) and leaf dry matter content (LDMC) are commonly used to reflect trees’ resource utilization strategies. In habitats with ample water and thermal conditions, trees increase SLA and decrease LDMC to reduce leaf construction costs and enhance their photosynthetic capacity, adopting a fast investment–fast return resource utilization strategy [32]. When SLA is high and LDMC is low, trees allocate more resources to aboveground parts (such as leaves and branches), aiding in rapid sunlight capture and increased photosynthesis [33]. High leaf nitrogen and phosphorus contents are often associated with high photosynthetic efficiency and rapid growth. In high nitrogen and phosphorus conditions, trees allocate more resources to aboveground parts to support rapid growth, thereby reducing BGBP [34]. However, some studies have found that plants need to maintain a balance between aboveground and belowground parts during growth [35]. In certain conditions, even with ample nitrogen and phosphorus nutrients, plants may still need to increase root biomass to support the rapidly growing aboveground parts, especially when aboveground parts are limited by environmental factors (such as light and moisture) [34].

This study, based on biomass data from 926 forests at 94 sites across China collected through field surveys and the literature from 2005 to 2020, investigates the patterns and controlling factors of forest BGBP along elevational gradients at a macro scale. To address these aforementioned issues, the following hypotheses are proposed: (1) BGBP exhibits a significant increasing trend with an elevational gradient. (2) Climatic factors predominantly drive the variation in BGBP along elevations, though soil nutrients and key leaf traits also play significant roles. (3) Climatic factors not only directly influence the elevational patterns of BGBP but also indirectly affect them by regulating soil nutrient availability and key leaf traits.

## 2. Materials and Methods

### 2.1. Experimental Sites and Sampling

To examine the spatial distribution of BGBP and its associated factors across elevational gradients, we selected 926 forest sites in China as our study areas. In this study, the dataset encompassing 926 forests from 94 locations comprised two primary sources: 628 forest datasets from 73 sites were gathered through the literature review, while the remaining 298 forest datasets from 21 sites were obtained from experiments conducted in this research. These sites were established between 2005 and 2020 (Figure 1A), with specific data sources detailed in Appendix A. The data extracted from the literature were processed and calculated according to consistent experimental methods. The data collected in this study are consistent with the sampling method given in the literature so as to ensure the uniformity of all samples. At each site, a minimum of four 30 × 30 m forest sample plots were chosen, each representing the typical zonal vegetation of the area. We recorded the latitude, longitude, elevation, and slope of each site for comprehensive analysis. Our study focused solely on tree species, as they constitute the majority of forest biomass. In each plot, all trees with a Diameter at Breast Height (DBH) over 1 cm were cataloged, allowing for the assessment of species richness, biomass, and stand density. The BGBP was calculated by dividing the BGB by the sum of AGB and BGB.

### 2.2. AGB and BGB Determination

#### 2.2.1. AGB Determination

In each plot, we selected healthy and mature tree species with a DBH close to the average value of the plot and sampled 10 individuals (DBH > 5 cm) from each tree species. The aboveground parts of these trees were categorized into bark, leaves, and branches. The trunk was segmented in 2 m intervals, with each section weighed to ascertain the trunk’s total fresh weight. For the branches and leaves, the average standard branch method was utilized to choose representative branches. These branches were weighed repeatedly to determine their fresh weight. Leaves were then carefully removed, placed in sealed bags with desiccants, and weighed subsequently. All three components—trunk, branches, and leaves—were transported to the laboratory and dried at 70 °C for 48 h. After drying, we measured the dry weight of each component, facilitating the calculation of the total AGB for each tree.

#### 2.2.2. BGB Determination

The entire root system of each sampled tree was meticulously excavated to a depth of 50 cm in the soil. We separated the roots and root clumps, carefully removing the surrounding soil. The fresh weight of the roots was then accurately measured. Selective sampling was conducted for root clumps weighing between 1000 and 2000 g and root sections between 500 and 1000 g, which were then transported to the laboratory. Although fine roots (< 5 mm) constituted a minor proportion of the total biomass and were challenging to harvest completely, diligent efforts were made to collect as many as possible, and the weight of all harvested roots was recorded [36]. To determine the BGB, all root samples were placed in a ventilated drying oven set at 85 °C until a consistent weight was achieved. The dry weight of these root samples was then used for subsequent calculations.

### 2.3. Plant Functional Traits and Environmental Factors

The leaf area (LA), specific leaf area (SLA), leaf nitrogen content (N_mass), leaf phosphorus content (P_mass), N_mass/P_mass ratio, and leaf dry matter content (LDMC) were determined. These leaf functional traits are crucial for describing and explaining plant survival strategies in adapting to diverse environmental conditions [37,38,39].

In each forest plot, leaves free of pests and diseases were collected from over 20 mature and well-developed trees of each species. These leaves were harvested from the outer branches of the tree crown using scissors. They were then placed between two sheets of moist filter paper and sealed in self-sealing bags for transportation to the laboratory. Upon arrival, the bags were stored in a refrigerated compartment. Prior to further analysis, the leaves were removed from their bags and incubated for 12 h in the dark at 5 °C to eliminate excess surface moisture. The leaves were then gently blotted with filter paper to remove any remaining water and weighed on an electronic balance to obtain their saturated fresh weight.

For the determination of dry weight, the leaves were thoroughly dried in an oven at 80 °C for 24 h and then weighed. LDMC was calculated by dividing the dry weight of the leaves (in mg) by their saturated fresh weight (in g).

To assess LA and SLA, a scanner (UNIS K3000C, Unisplendour Corporation Limited, Beijing, China) was used to scan the leaves, and Image-J software (https://imagej.en.softonic.com/, Version 1.8.0, accessed on 1 Jul 2023) was employed to calculate their area. For conifer species, each bundle of needles was treated as a cylinder, estimating the SLA by considering half of the total surface area. This method yielded results for broadleaf and other flat leaves consistent with those obtained by measuring individual leaves.

For leaf nutrient analysis, the Kjeldahl method was used to determine the total nitrogen content, while the Mo-sb colorimetry method was utilized for measuring the total phosphorus content in the leaves [40].

Previous studies on leaf traits have primarily focused on species-level measurements [37]. However, there has been limited exploration into how competitive asymmetry associated with functional traits varies among species [40]. To address this, the community-weighted mean value (CWM) was employed to represent the average trait value of each forest. This approach integrates the trait values of individual species, weighted by their relative abundance within the community. By using CWM, we can gain a more comprehensive understanding of the functional trait composition at the community level, which is crucial for interpreting ecosystem processes and dynamics. This method provides a more holistic view of forest ecosystems, allowing us to better understand how different species interact and contribute to overall forest structure and function. The plant functional trait data used in this study came from the dataset published by Zhang et al. [41], which has been widely used in the study of plant functional traits on a macro scale.
$$\mathrm{CWM}=\sum_{i=1}^{S}D_{i}\;\;\times {\mathrm{Trait}}_{i}\;\;\;\;\;\;$$
where CWM represents the weighted functional traits, D_i_ represents the abundance of dominant tree species, and Trait_i_ represents the selected functional traits [42]. In subsequent research, we discovered that only leaf nitrogen content and leaf phosphorus content showed a significant correlation with BGBP (*p* < 0.05). Consequently, leaf nitrogen and phosphorus contents were used to represent the functional trait of the community.

### 2.4. Environmental Data

The Mean Annual Temperature (MAT), Mean Hottest Monthly Temperature (MAHT), Mean Coldest Monthly Temperature (MCMT), and Mean Annual Precipitation (MAP) data were extracted from the WorldClim (https://www.worldclim.org/, accessed on 1 July 2023) global climate database with a spatial resolution of 1 km [43]. The Annual Sunlight Duration(ASD) and Mean Annual Evaporation (MAE) data were extracted from the China Meteorological Data Service Centre (https://data.cma.cn/site/index.html, accessed on 1 July 2023) with a spatial resolution of 1 km. Soil pH, nitrogen (N), and phosphorus (P) values for the top 30 cm layer of soil were acquired from [41] and https://www.osgeo.cn/data/wc137 with a spatial resolution of 250 m (accessed on 1 July 2023).

### 2.5. Data Analysis

Before data analysis, we standardized the data, and subsequent data analysis used the same sample size, that is, all data in Appendix A. To examine the contribution of both abiotic and biotic factors to the spatial variation in BGBP and to account for the impact of random effects on biomass allocation, we utilized a linear mixed-effects model. The goodness of fit for these models was evaluated using the marginal R^2^ value [44]. Climatic factors included the Mean Annual Temperature (MAT), Mean Annual Precipitation (MAP), Mean Hottest Monthly Temperature (MAHT), Mean Coldest Monthly Temperature (MCMT), and Annual Sunshine Duration (ASD). Soil factors comprised soil nitrogen content (soil N), soil phosphorus content (soil P), and Soil pH. Community functional traits included the leaf nitrogen content and leaf phosphorus content. A linear mixed-effects analysis was conducted using the R package “lme4” [45]. This approach allowed for a comprehensive assessment of the relative influences of these varied factors on BGBP across different forest ecosystems.

Due to the numerous environmental factors we considered, there was a significant correlation between these factors (Figure 5), which might cause multicollinearity. Therefore, we established a stepwise regression model and selected the optimal model based on the lowest AICc value (AIC values corrected for sample size), thereby eliminating irrelevant variable factors.

Variance decomposition methods were used to quantify the explanatory power of various climatic factors, soil factors, and key leaf traits on the spatial variability of BGBP. Variance decomposition analysis was conducted using the R (Version 4.2.2) software package “rdacca.hp” [46]. Furthermore, the independent contribution of each potential influencing factor to the spatial variation in key leaf nutrient traits was explored using the machine learning method of boosted regression trees, with significance testing at the 0.05 level. This analysis was carried out using the package “randomForest” [47]. These methods provide a robust framework for assessing the relative impacts of these factors on BGBP, offering a nuanced understanding of how various environmental and biological factors interplay to shape forest biomass allocation.

The use of structural equation modeling (SEM) explored the pathways through which climatic factors, soil nutrient factors, and functional traits influence the BGBP [48]. All observed variables were initially grouped into composite variables and incorporated into the SEMs. In order to verify the reliability of the relationship between key ecological factors and BGBP, we utilized piecewiseSEM to elucidate the random effects of sampling points and provide both the “marginal” and “conditional” contributions of environmental predictive factors. These analyses were conducted using the “piecewiseSEM,” “nlme,” and “lme4” packages. The goodness of fit of the modeling results was assessed using Fisher’s C-test. Based on a significance level of *p* < 0.05 and satisfactory model goodness (0 ≤ Fisher’s C/df ≤ 2 and 0.05 < *p* ≤ 1.00), the model was incrementally adjusted through a stepwise modification process [49].

## 3. Results

BGBP exhibits significant variability at a macro scale, showing a marked increasing trend along elevational gradients (R^2^ = 0.15, *p* < 0.001, Figure 1). BGBP significantly decreases with increases in temperature factors (MAT, MACT, MAHT (R^2^ = 0.14, *p* < 0.001, Figure 2A; R^2^ = 0.10, *p* < 0.001, Figure 2B; R^2^ = 0.05, *p* < 0.001, Figure 2C)) and the precipitation factor (MAP) (R^2^ = 0.47, *p* < 0.001, Figure 2D), and significantly increases with increases in MAE (R^2^ = 0.09, *p* < 0.001, Figure 2F), and with increases in Annual Sunshine Duration (ASD) (R^2^ = 0.53, *p* < 0.001, Figure 2E). Among all climatic factors, MAP and ASD provide the best explanation for the spatial variability of BGBP (R^2^ = 0.47, *p* < 0.001, Figure 2D; R^2^ = 0.53, *p* < 0.001, Figure 2E).

BGBP significantly decreases with an increase in soil phosphorus content (R^2^ = 0.20, *p* < 0.001, Figure 3B) and significantly increases with an increase in soil pH (R^2^ = 0.44, *p* < 0.001, Figure 3C), while changes in soil nitrogen content are not significant (*p* > 0.05). Soil pH (R^2^ = 0.44) explains more of the spatial variability in BGBP than soil phosphorus content (R^2^ = 0.20) and soil nitrogen content (R^2^ < 0.01). Key leaf traits (LN and LP) are significantly positively correlated with BGBP (*p* < 0.001), while correlations with other key leaf traits are not significant (*p* > 0.05) (Figure 4).

The potential influencing factors of BGBP show significant correlations with each other (*p* < 0.05) (Figure 5), and variance decomposition results indicate that climatic factors (R^2^ = 0.46) have the strongest explanatory power for the elevational variation patterns of BGBP, with soil factors (R^2^ = 0.10) and key leaf traits (R^2^ = 0.08) also playing significant roles (Figure 6). The results from the boosted regression tree model show that soil pH has the strongest independent explanatory power for the spatial variation in BGBP, with key leaf traits (LN and LP) having weaker explanatory power (Figure 7).

Structural equation modeling results indicate that elevation not only directly affects BGBP but also indirectly influences it through effects on climatic conditions, soil nutrient availability, and key leaf traits, with direct impacts being more pronounced than indirect impacts (Figure 8).

## 4. Discussion

Plants at different elevations develop corresponding survival strategies to adapt to complex living environments [50]. Research findings indicate a significant increase in forest BGBP with increasing elevation. High-elevation areas are generally characterized by harsh environments, imposing greater survival pressures on plants. Consequently, plants adopt conservative resource utilization strategies, allocating a larger proportion of biomass to belowground parts [51]. Additionally, forests at different elevational heights have varying species compositions [52]. In high-elevation areas, species capable of effectively adapting to soil and climatic conditions tend to develop more robust root systems, leading to higher BGBP [2].

BGBP significantly decreases with increases in temperature and precipitation. According to the functional balance hypothesis, plants adjust their biomass allocation based on environmental conditions [53]. Under warm and moist conditions, plants may prioritize the growth of aboveground parts (such as leaves and branches) to utilize abundant moisture and higher photosynthetic potential, resulting in a decreased proportion of belowground biomass [54,55]. As temperature increases, plant metabolism often accelerates [5], meaning they can conduct photosynthesis and other physiological processes quicker, potentially increasing the growth rate of aboveground parts [56]. By contrast, the growth of belowground parts may not accelerate at the same rate, leading to a decrease in the proportion of belowground biomass.

Increased precipitation generally implies improved soil moisture conditions. Under conditions of ample moisture, plants may not need to develop extensive root systems to acquire water, thus potentially reducing investment in belowground biomass and allocating more resources to the growth of aboveground parts, like leaves and stems [57]. Forest BGBP significantly increases with longer daylight duration. Increased daylight may lead to higher transpiration, thus increasing plants’ water demand [58]. To efficiently absorb and utilize water, plants develop larger and deeper root systems, causing an increase in the BGBP [35]. According to the functional balance theory, plants adjust their resource allocation based on the limiting factors for resource acquisition [53]. In conditions of ample sunlight, when photosynthesis is no longer a limiting factor, plants may allocate more resources (such as photosynthetic products) to the growth and development of roots to more effectively absorb water and nutrients from the soil, especially in nutrient-limited environments [59].

BGBP significantly decreases with increased phosphorus content. In natural environments, phosphorus is often one of the key nutrients limiting plant growth. In soils with low phosphorus levels, plants tend to increase their root biomass and the extent of their root systems to enhance phosphorus absorption efficiency, enabling the more effective exploration of phosphorus in the soil [60]. As soil phosphorus levels increase and this limitation is reduced, plants no longer need to invest heavily in belowground growth. With higher soil phosphorus levels, plants can more easily obtain the phosphorus they need from the soil, thus reducing their reliance on extensive root systems [61]. Forests in alkaline soils tend to have a higher BGBP, as alkaline conditions often affect the availability of certain key nutrients, especially micronutrients like iron and manganese [62,63]. The solubility of these elements decreases under alkaline conditions, making them harder for plants to absorb. To overcome this nutrient limitation, plants might increase root biomass to enhance their acquisition of these micronutrients [64]. Alkaline soils may lead to reduced water use efficiency, as high pH values can affect the soil structure and water-holding capacity [65]. In order to absorb and use water more efficiently, plants may develop larger and deeper root systems, increasing belowground biomass [66].

Our study also found that the soil nitrogen content does not significantly influence the spatial variability of BGBP. This is primarily because plants typically adjust their biomass allocation based on the availability of environmental resources. When soil nitrogen levels are already sufficient to support plant growth, further increases in nitrogen may not significantly change the allocation ratio of aboveground-to-belowground biomass [34]. Different plants have varying nitrogen use efficiencies, and some can maintain high growth rates even in low nitrogen environments, so increased nitrogen levels might not significantly affect the biomass allocation of these plants [67].

In addition to abiotic factors like climate and soil, key leaf traits in forests also play a crucial role in driving the spatial variability of BGBP [33]. Our study found that BGBP significantly increases with the increase in leaf nitrogen and phosphorus content. Nitrogen and phosphorus are two key nutrients limiting plant growth, and they are essential for the development and expansion of root systems [68]. An increase in leaf nitrogen and phosphorus content may lead to more energy and resources being allocated to the growth and development of root systems, thereby increasing the BGBP [69].

Climatic factors have a stronger explanatory power on the spatial variability of forest BGBP than soil factors and key leaf traits. This is because climatic conditions, such as temperature, precipitation, sunlight, and humidity, not only directly affect plant growth cycles, photosynthesis efficiency, and water use efficiency but also play a decisive role in the formation and accumulation of belowground biomass [70]. For instance, sufficient rainfall and suitable temperatures can promote root growth and expansion, thereby increasing belowground biomass [71]. The distribution and seasonal fluctuations of climatic factors in different geographical areas significantly affect plant growth patterns and energy allocation strategies [35]. For example, in water-limited environments, plants may be more inclined to develop their root systems to improve water acquisition, directly impacting BGBP [17]. Climatic conditions also indirectly affect plant root development by influencing the physical and chemical properties of the soil and the activity of soil microorganisms [72]. For example, lower temperatures and humidity may limit microbial activity, affecting nutrient cycling and, consequently, influencing root growth [15].

Elevation not only directly affects BGBP but also indirectly influences it through its impact on other biotic and abiotic factors (Appendix A), with direct effects being more pronounced than indirect ones. Elevation affects plant growth cycles and metabolic rates [73]. Lower temperatures at higher elevations may slow root growth and soil microbial activity, directly impacting the accumulation of belowground biomass [71]. An increase in elevation is often accompanied by changes in soil nutrient availability [74,75]. Cold temperatures may slow down the decomposition of organic matter, affecting nutrient release and cycling [7]. However, these effects are often strongly influenced by direct climatic factors. The direct impact of elevation is typically more significant because it directly alters the environmental conditions for plant growth, such as temperature and moisture, which are key factors influencing root development and the accumulation of belowground biomass. Although indirect effects are also important, they are generally controlled and modulated by direct influencing factors. In practical fieldwork, diverse geographical conditions and soil texture variations make it challenging to uniformly obtain root biomass at a depth of 1 m. To align our field data collection methodology with that of the literature, we opted for a soil depth of 50 cm. Recognizing this approach underestimates underground biomass, and we acknowledge it as a limitation of our study and plan to address it in future research.

## 5. Conclusions

The results show that elevation had a significant effect on above–belowground biomass distribution. Elevation not only directly affects forest biomass allocation but also indirectly affects BGBP through other biological and abiotic factors, and the direct effect is greater than the indirect effect. Compared with other influencing factors, climate factors determine the elevation gradient pattern of forest above–belowground biomass allocation distribution. The results of our study enrich our understanding of the effect of elevation gradient change on forest growth and provide references for the formulation of forest management strategies at different elevations in the future.

## Figures and Tables

**Figure 1 plants-13-00674-f001:**
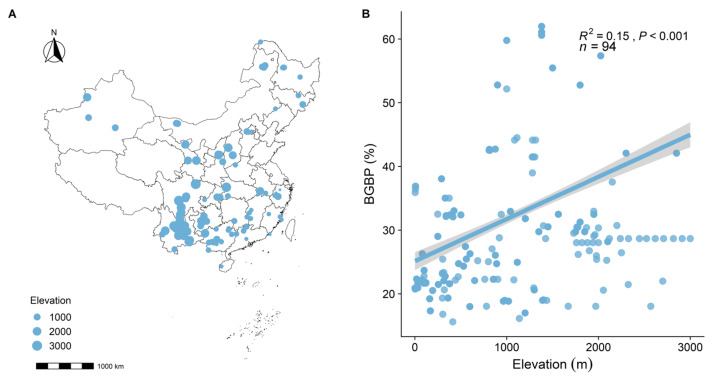
Distribution of research sites and the variation in belowground biomass proportion (BGBP) along the elevational gradient. (**A**) Geographic location and elevation distribution map of forest plots in this study; (**B**) General linear correlation analysis between BGBP and elevation. R^2^ represents the goodness of fit of the model, and the *p*-value indicates the level of significance.

**Figure 2 plants-13-00674-f002:**
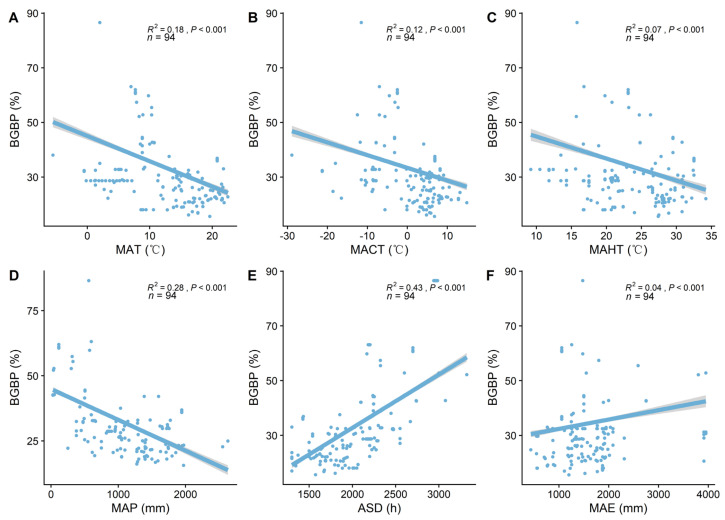
The linear relationship between climatic factors and belowground biomass proportion (BGBP). R^2^ indicates the model’s goodness of fit, and the *p*−value represents significance. Climatic factors include the following: (**A**) Mean Annual Temperature (MAT); (**B**) Mean Coldest Monthly Temperature (MCMT); (**C**) Mean Hottest Monthly Temperature (MAHT); (**D**) Mean Annual Precipitation (MAP); (**E**) Annual Sunlight Duration (ASD); (**F**) Mean Annual Evaporation (MAE).

**Figure 3 plants-13-00674-f003:**
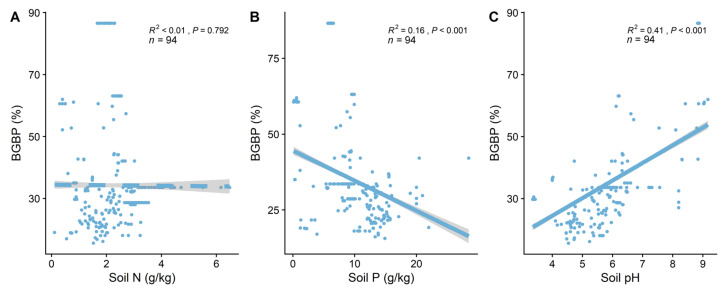
The linear relationship between soil factors and BGBP. R^2^ indicates the model’s goodness of fit, and the *p*−value represents significance. The soil factors include the following: (**A**) soil total nitrogen (N) content (soil N); (**B**) soil available phosphorus (P) content (soil P); and (**C**) soil pH.

**Figure 4 plants-13-00674-f004:**
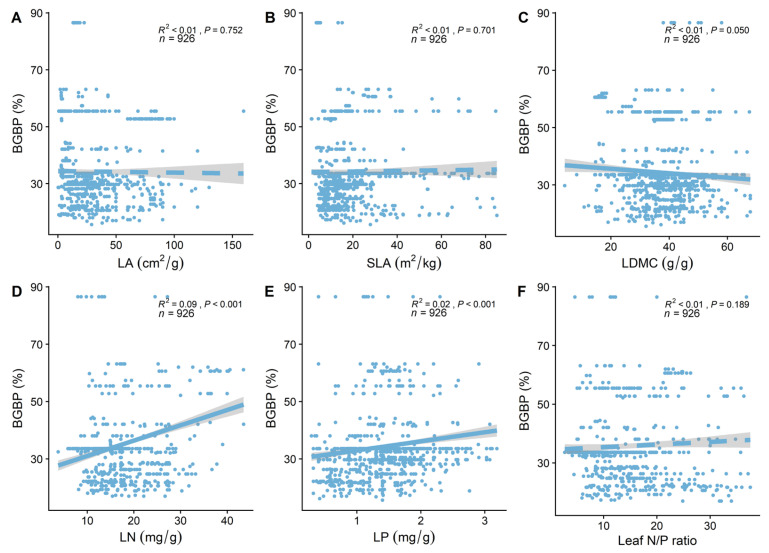
The linear relationship between leaf functional traits and belowground biomass proportion (BGBP). R^2^ indicates the model’s goodness of fit, and the *p*−value represents significance. The leaf functional traits include the following: (**A**) leaf area (LA); (**B**) specific leaf area (SLA); (**C**) leaf dry matter content (LDMC); (**D**) leaf nitrogen content (LN); (**E**) leaf phosphorus content (LP); (**F**) and leaf nitrogen to phosphorus ratio (N/P ratio).

**Figure 5 plants-13-00674-f005:**
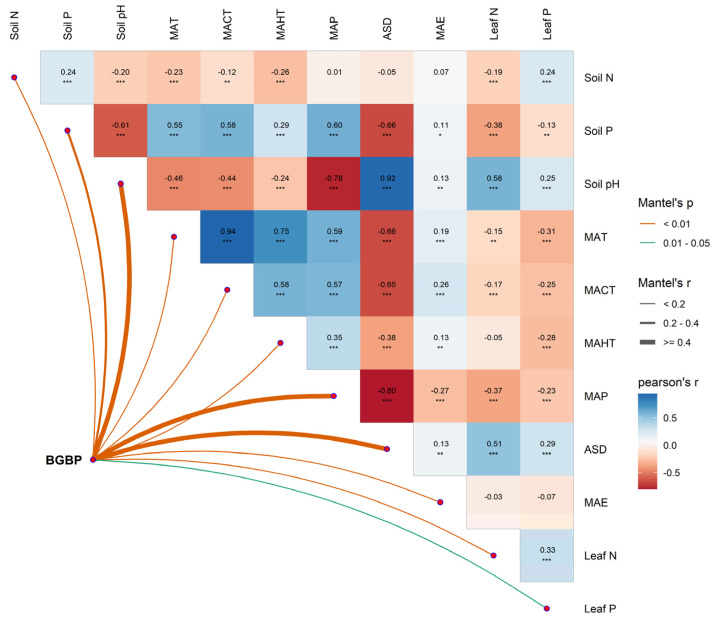
Multivariate correlation analysis of the potential influencing factors of BGBP. The influencing factors include climatic factors (Mean Annual Temperature [MAT]; Mean Hottest Monthly Temperature [MAHT]; Mean Coldest Monthly Temperature [MCMT]; Mean Annual Precipitation [MAP]; Annual Sunlight Duration [ASD]; Mean Annual Evaporation [MAE]), soil factors (soil total nitrogen [N] content [soil N]; soil available phosphorus [P] content [soil P]; soil pH), and leaf functional traits (leaf nitrogen [LN]; leaf phosphorus [LP]). * *p* < 0.05; ** *p* < 0.01; *** *p* < 0.001.

**Figure 6 plants-13-00674-f006:**
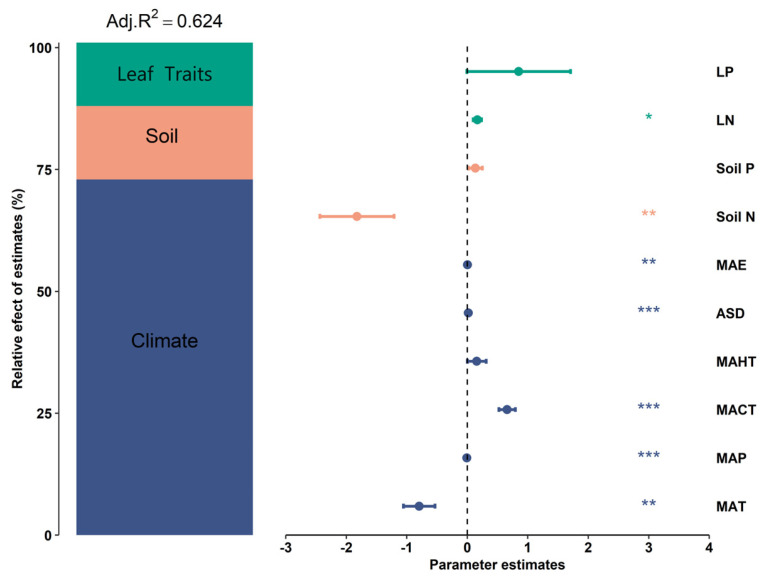
Impact of potential influencing factors on BGBP. Mean parameter estimates (standardized regression coefficients) of model predictors, associated 95% confidence intervals, and relative importance of each factor are represented as the percentage of explained variance. Adjusted R^2^ for the average model and *p*−values for each predictor are as follows: * *p* < 0.05; ** *p* < 0.01; *** *p* < 0.001.

**Figure 7 plants-13-00674-f007:**
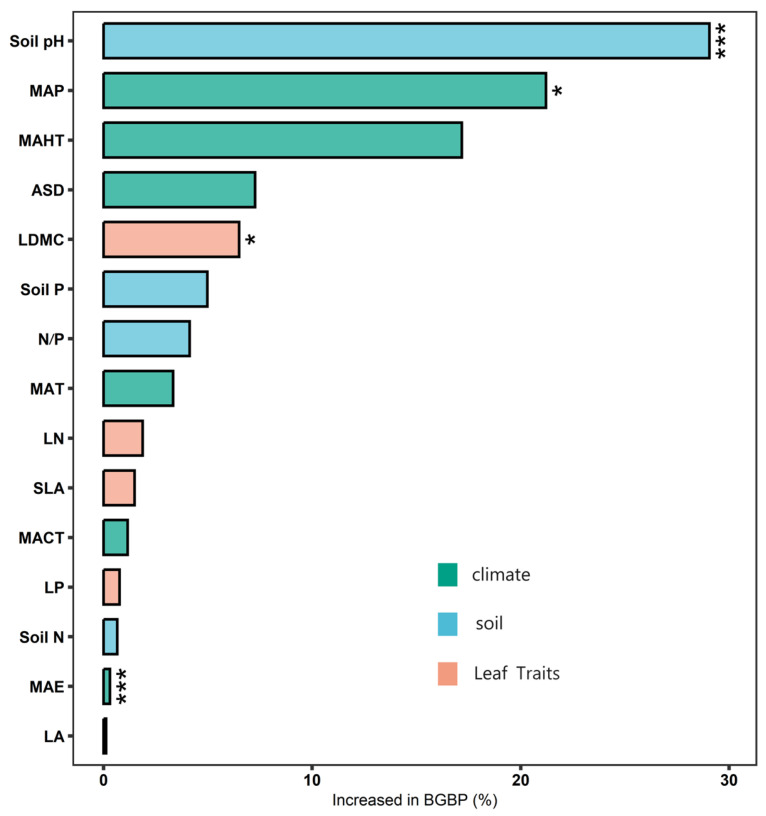
Independent contributions of three categories of factors—climatic factors, soil factors, and leaf traits—for the BGBP of forests. Pink represents leaf traits, blue represents soil factors, and green represents climatic factors. Climatic factors include Mean Annual Temperature [MAT], Mean Hottest Monthly Temperature [MAHT], Mean Coldest Monthly Temperature [MCMT], Mean Annual Precipitation [MAP], Annual Sunlight Duration [ASD], and Mean Annual Evaporation [MAE]. Soil factors include soil total nitrogen (N) content [Soil N], soil available phosphorus (P) content [Soil P], and soil pH. Leaf traits include leaf nitrogen [LN] and leaf phosphorus [LP]. * *p* < 0.05; *** *p* < 0.001.

**Figure 8 plants-13-00674-f008:**
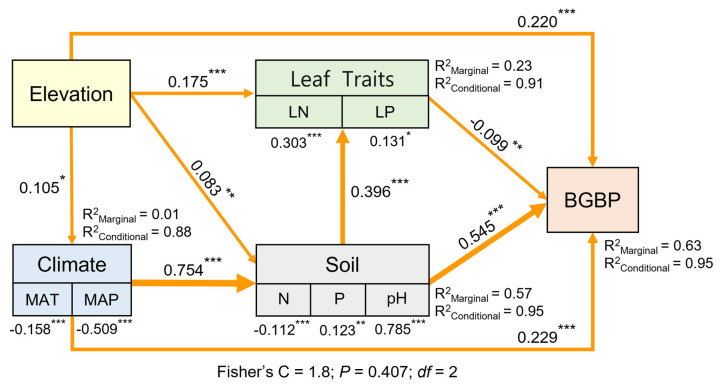
Impact pathways of elevational gradient, climatic factors, soil factors, and key leaf traits on BGBP. The width of the arrows is proportional to the level of significance, with numbers next to the paths representing standardized SEM coefficients. Asterisks indicate significant differences (*** *p* < 0.001; ** *p* < 0.01; * *p* < 0.05). R^2^ indicates the goodness of fit for the Generalized Additive Model.

## Data Availability

Data can be seen in Appendix A.

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
