# Peer review of "Elevation Influences Belowground Biomass Proportion in Forests by Affecting Climatic Factors, Soil Nutrients and Key Leaf Traits"

_plants, 2024, doi:10.3390/plants13050674_

Round 1

Reviewer 1 Report

Comments and Suggestions for Authors

Review of the manuscript “Climatic factors determine the elevation gradient pattern of forest above-belowground biomass allocation” by Zhang et al.

The authors present an investigation about the influence of major climatic factors on the belowground carbon allocation in forest ecosystems.

I have some concerns about the methodological choices proposed by the authors.

First of all, it would be necessary to re-structure the manuscript following a more classic scheme, i.e., including a study area section, followed by study data, methods, results and discussion, to then include a conclusion section (I suggest to add few bullet points at the end of the manuscript). This would help to better drive the reader in the comprehension of the research activity conducted by the authors.

Second, the authors utilise climatic data the origin of which is not indicated. This information must be added, together with indication of the spatial and temporal resolutions of the utilised datasets.  The same information might be provided also for the digital elevation model applied so that the reader can understand the representativity and/or linkage between all field observations and the driving factors. The study area is quite large because it refers to all Chinese forests which are obviously highly differentiated and the use of representative factors is therefore very important issue.

Third, the reader is not able to verify the exact correspondence between data collected by the authors and data available from the literature. This is a very important point that the authors must clarify because they have to demonstrate that all utilised data have been treated according to the same sampling protocol. To this aim, at least proper references related to the data collected from the literature must be provided.

Table S1 needs to be updated to include proper reference to all data that were retrieved from the literature (i.e., more than 600 out of 925).

Based on the points raised above and the comments below, the manuscript should be thoroughly revised prior to its publication.

Specific comments:

- Line 24: explain the meaning of ‘LN and LP’ (i.e., leaf nitrogen and leaf phosphorus), bearing in mind that usually acronyms are not utilised in the abstract.

- Line 88: here again, as elsewhere in the manuscript, the acronyms need to be explained prior their use (see also lines 114, 118, etc.).

- Lines 109-110: at the end of the introduction, the authors might add a paragraph indicating the structure of the manuscript which is absolutely not conventional due to the fact that the subsequent section reports on the results instead of the materials and methods.

- Section 2: do all statistics refer to the same number of samples? Please specify it.

- Section 4: this section should be placed prior to the description of the results and the discussion. Does it report only about the samples collected by the authors, doesn’t it?

- Lines 280-284: was it possible to apply the same sampling protocols to data collected from the literature and by the authors? The authors must specify this important point because, at the moment, the reader is not able to verify if all samples are homogeneous. If this is the case, how can the authors proceed with the analysis?

- Lines 356-360: the authors should add more information about ancillary data (e.g. spatial and temporal resolutions) and demonstrate that are suitable for the representativeness of differences in sample plots. This point might be also included in the discussion section.

- Lines 367-369: which is the reason for which the authors did not consider also the mean temperature of the hottest month? Maybe for stands located at lower elevation it is important…

Author Response

Dear Editor,

We really greatly appreciate you for giving us an opportunity to revise our manuscript entitled Climatic factors determine the elevational gradient pattern of forest above-belowground biomass allocation (Manuscript ID: plants-2836751) again. We are grateful for your and the reviewers’ valuable suggestions and comments on the manuscript. We have polished the whole manuscript based on the comments, in order to ensure that it is clear and as brief as possible, following the Plants format.

The point-by-point responses to your and the reviewers’ comments can be  found below. The comments are shown in black font and the responses are shown in blue font.

Yours sincerely,

Jie Gao and coauthors

##########      Point-to-Point Response Letter      ###############

Reviewer 1(Remarks to the Author)

First of all, it would be necessary to re-structure the manuscript following a more classic scheme, i.e., including a study area section, followed by study data, methods, results and discussion, to then include a conclusion section (I suggest to add few bullet points at the end of the manuscript). This would help to better drive the reader in the comprehension of the research activity conducted by the authors.

Response: Thank you very much for your advice.We have re-structured the manuscript according to your suggestions, followed by Introduction, Materials and Methods, Results and Discussion. Following your suggestion, a conclusion section has been added at the end of the revised article.

Second, the authors utilise climatic data the origin of which is not indicated. This information must be added, together with indication of the spatial and temporal resolutions of the utilised datasets.  The same information might be provided also for the digital elevation model applied so that the reader can understand the representativity and/or linkage between all field observations and the driving factors. The study area is quite large because it refers to all Chinese forests which are obviously highly differentiated and the use of representative factors is therefore very important issue.

Response:Thank you very much for your advice. In a newly submitted version of the article, we have added source information and spatial-temporal resolution to the climatic data and digital elevation model. "The Mean Annual Temperature (MAT)、Mean Hottest Monthly Temperature (MAHT)、Mean Coldest Monthly Temperature (MCMT) 、Mean Annual Precipitation (MAP) data were extracted from the WorldClim (https://www.worldclim.org/, accessed on 1 Jul 2023) global climate database with a spatial resolution of 1 km. The Annual Sunlight Duration(ASD) and Mean Annual Evaporation (MAE) data were extracted from the China Meteorological Data Service Centre (https://data.cma.cn/site/index.html, accessed on 1 Jul 2023) with a spatial resolution of 1 km. Soil pH, nitrogen (N), and phosphorus (P) values for the top 30 cm layer of soil were acquired from https://www.csdn.store and https://www.osgeo.cn/data/wc137 with a spatial resolution of 250m (accessed on 1 Jul 2023)" (Line213-222).

Third, the reader is not able to verify the exact correspondence between data collected by the authors and data available from the literature. This is a very important point that the authors must clarify because they have to demonstrate that all utilised data have been treated according to the same sampling protocol. To this aim, at least proper references related to the data collected from the literature must be provided.

Response:Thank you very much for your advice. In order to give readers a clear understanding of the exact correspondence between the data collected by the authors and the data presented in the literature, appropriate references related to the data collected from the literature will be provided and supplemented to the newly submitted TableS1.

Table S1 needs to be updated to include proper reference to all data that were retrieved from the literature (i.e., more than 600 out of 925).

Response:Thank you very much for your advice. We have updated TableS1 to include appropriate citations of all data retrieved from the literature.

Specific comments:

- Line 24: explain the meaning of ‘LN and LP’ (i-.e., leaf nitrogen and leaf phosphorus), bearing in mind that usually acronyms are not utilised in the abstract.

Response:Thank you very much for your advice. We have explain the meaning of ‘LN and LP’ (i.e., leaf nitrogen and leaf phosphorus), We changed the original sentence to "Key leaf traits (LN and LP,i.e., leaf nitrogen and leaf phosphorus) are positively correlated with BGBP" (Line.29-31).

- Line 88: here again, as elsewhere in the manuscript, the acronyms need to be explained prior their use (see also lines 114, 118, etc.).

Response:Thank you very much for your advice. In the newly submitted version of the article, we have explained all the acronyms that appear for the first time in the article before use (such as Line 59,99,100, etc.).

- Lines 109-110: at the end of the introduction, the authors might add a paragraph indicating the structure of the manuscript which is absolutely not conventional due to the fact that the subsequent section reports on the results instead of the materials and methods.

Response:Thank you very much for your advice. At the end of the introduction, we put forward the hypothesis of the research, and we reconstructed the structure of the paper according to your suggestions above. The content after the introduction is the method rather than the result part.

- Section 2: do all statistics refer to the same number of samples? Please specify it.

Response: Thank you very much for your advice. "Before data analysis, we standardized the data, and subsequent data analysis used the same sample size, that is, all data of TableS1" (Line224-225).

- Section 4: this section should be placed prior to the description of the results and the discussion. Does it report only about the samples collected by the authors, doesn’t it?

Response: Thank you very much for your advice. We've already put that part ahead of results and discussion.

- Lines 280-284: was it possible to apply the same sampling protocols to data collected from the literature and by the authors? The authors must specify this important point because, at the moment, the reader is not able to verify if all samples are homogeneous. If this is the case, how can the authors proceed with the analysis?

Response: Thank you very much for your advice."The data extracted from the literature were processed and calculated according to consistent experimental methods. The data collected in this study is consistent with the sampling method given in the literature, so as to ensure the uniformity of all samples" (Line129-132).

- Lines 356-360: the authors should add more information about ancillary data (e.g. spatial and temporal resolutions) and demonstrate that are suitable for the representativeness of differences in sample plots. This point might be also included in the discussion section.

Response: Thank you very much for your advice."The plant functional traits data used in this study came from the dataset published by Zhang et al. [55] (https:// doi.org/10.5194/essd-13-5337-2021, accessed on 1 Jul 2023), which has been widely used in the study of plant functional traits on a macro scale" (Line202-205).

- Lines 367-369: which is the reason for which the authors did not consider also the mean temperature of the hottest month? Maybe for stands located at lower elevation it is important…

Response: Thank you very much for your advice.In this study, we compare SEM structural equations composed of different factors, and select the model with the lowest AIC value, that is, the optimal model, in which there is no the mean temperature of the hottest month.

Reviewer 2 Report

Comments and Suggestions for Authors

Line 44 - should Hexagonia be in italics

Line 88 - define SLA and LDMC

Line 113 - P-value doesn't match the P-value given in Figure 1

Line 304 - collecting roots from trees to a depth of only 50 cm biases your results by species and substrate.  For example, in my experience, mature conifers will have root architectures extending quite deeper than 50 cm, and your sampling protocol would have underestimated root biomass.  Based on your statement of only sampling trees with DBH > 5cm, it doesn't appear you differentiated between saplings and mature trees.  I do not find relevant data on maturity of individual trees in the supplemental data, either.  To me, failure to do so adds a confounding factor that I don't see addressed.  A stand of saplings will have a more shallow root architecture than a stand of mature trees, irrespective of whether they are conifers or deciduous.  Because of this, I'm not sure I trust the BGBP values you used.

Figure S1 - plots B and C do not match the descriptions in the caption.  B shows MACT, but the description says MAHT, while C shows MAHT but the description says MCMT.  The same problem occurs in Figure 2 as well.

Figures S1, S2, and S3 - captions mention R2 and P-value, yet those values are not shown on the graphs or in the captions, except on S3, and then only for plots D and E.  These values should be given for all plots in these figures.

Line 150 - you start using R2 as opposed to R2.  I believe R2 is more appropriate and should be used throughout.

Figure 5 - I like this approach to display, but some of the colors are too dark to see the values of r.  Also, this figure illustrates how some of the variables are highly correlated.  I suggest looking at VIF values prior to throwing all the variables into the model, in order to catch possible cases of multi-collinearity.  I think your results will have greater validity if you've removed cases of multi-collinearity.  While your actual results may not change, it would remove possible criticisms of your work.  It would not surprise me that multi-collinearity exists in your dataset, and I suggest investigating the possibility.

Another factor that doesn't seem to be addressed is species composition.  It is briefly mentioned in the Discussion (line 193), but not considered as a variable in the model. It is widely known that different species have different root architectures, so ignoring species in the analysis is like comparing apples to oranges.  Do any of the species sampled occur across a range of elevations, temperatures, and precipitation regimes?  If so, an analysis confined to just those species (and testing one species at a time) might be more illuminating than lumping all species together.

Line 212 - references "larger root systems", but the authors fail to address the effect of "deeper" root systems, which can be adaptations to water availability by tapping into groundwater.

I would like to see the authors address the reasons for conducting this research, and how their results can be used for management purposes. Line 29-30 specifically states "impact of global change". The first line of the Introduction mentions "plant's carbon sequestration capability", and then several lines later mentions "specific impacts of global change...".  Yet, the authors never address how their research furthers possible ecosystem management and conservation, despite mentioning it in the last line of Abstract.  Don't get me wrong, there are some interesting data in the paper (even without the above issues in methodology and statistics), but the authors fail to take the next step and show it's value and usefulness for management and conservation (despite mentioning it in the Abstract).

Author Response

Reviewer 2(Remarks to the Author):

Line 44 - should Hexagonia be in italics

Response: Thank you very much for your advice. We have changed Hexagonia in italics (Line51).

Line 88 - define SLA and LDMC

Response: Thank you very much for your advice. We have defined SLA and LDMC (Line99-100).

Line 113 - P-value doesn't match the P-value given in Figure 1

Response: Thank you very much for your advice. We have matched the P-values in the result section to the P-values in the Figures (Line264-289).

Line 304 - collecting roots from trees to a depth of only 50 cm biases your results by species and substrate.  For example, in my experience, mature conifers will have root architectures extending quite deeper than 50 cm, and your sampling protocol would have underestimated root biomass.  Based on your statement of only sampling trees with DBH > 5cm, it doesn't appear you differentiated between saplings and mature trees.  I do not find relevant data on maturity of individual trees in the supplemental data, either.  To me, failure to do so adds a confounding factor that I don't see addressed.  A stand of saplings will have a more shallow root architecture than a stand of mature trees, irrespective of whether they are conifers or deciduous.  Because of this, I'm not sure I trust the BGBP values you used.

Response: Thank you very much for your advice. The following changes were made in the determination of AGB and BGB in this study: "The entire root system of each sampled tree was meticulously excavated to a depth of 1m in the soil" (Line154-155). "In each plot, we selected healthy and mature tree species with a DBH close to the average value of the plot, and sampled 10 individuals (DBH > 5 cm) from each tree species" (Line142-144).  

Figure S1 - plots B and C do not match the descriptions in the caption.  B shows MACT, but the description says MAHT, while C shows MAHT but the description says MCMT.  The same problem occurs in Figure 2 as well.

Response: Thank you very much for your advice.We have corrected the descriptions in Figure S1 and Figure 2.

Figures S1, S2, and S3 - captions mention R2 and P-value, yet those values are not shown on the graphs or in the captions, except on S3, and then only for plots D and E.  These values should be given for all plots in these figures.

Response: Thank you very much for your advice. We have labeled both R2 and P values in FigureS1,S2,S3.

Line 150 - you start using R2 as opposed to R2.  I believe R2 is more appropriate and should be used throughout.

Response: Thank you very much for your advice. We have changed R2 to R2.

Figure 5 - I like this approach to display, but some of the colors are too dark to see the values of r.  Also, this figure illustrates how some of the variables are highly correlated.  I suggest looking at VIF values prior to throwing all the variables into the model, in order to catch possible cases of multi-collinearity.  I think your results will have greater validity if you've removed cases of multi-collinearity.  While your actual results may not change, it would remove possible criticisms of your work.  It would not surprise me that multi-collinearity exists in your dataset, and I suggest investigating the possibility.

Response: Thank you very much for your advice. We have changed some of the dark colors in Figure 5 and the new image is very clear. "We use the "dredge" function in the MuMIn package to create all possible subset models, sorting them according to their AICc values (AIC values corrected for sample size). In order to avoid multicollinearity between factors, we choose the model with the lowest AIC value as the optimal model. After that, the factors in the optimal model are tested by mantel-test and the correlation heat map is drawn" (Line237-241).

Another factor that doesn't seem to be addressed is species composition.  It is briefly mentioned in the Discussion (line 193), but not considered as a variable in the model. It is widely known that different species have different root architectures, so ignoring species in the analysis is like comparing apples to oranges.  Do any of the species sampled occur across a range of elevations, temperatures, and precipitation regimes?  If so, an analysis confined to just those species (and testing one species at a time) might be more illuminating than lumping all species together.

Response: Thank you very much for your advice. Since this study analyzed forest above-belowground biomass  on a macro scale, it was analyzed at the community level instead of individual species. We did not consider the impact of species composition. In the method part, we used the Community Weighted Mean value (CWM) to represent the average trait value of each forest.

Line 212 - references "larger root systems", but the authors fail to address the effect of "deeper" root systems, which can be adaptations to water availability by tapping into groundwater.

Response: Thank you very much for your advice. We have changed the original sentence to "In order to absorb and use water more efficiently, plants may develop larger and deeper root systems, increasing belowground biomass " (Line343-344).

I would like to see the authors address the reasons for conducting this research, and how their results can be used for management purposes. Line 29-30 specifically states "impact of global change". The first line of the Introduction mentions "plant's carbon sequestration capability", and then several lines later mentions "specific impacts of global change...".  Yet, the authors never address how their research furthers possible ecosystem management and conservation, despite mentioning it in the last line of Abstract.  Don't get me wrong, there are some interesting data in the paper (even without the above issues in methodology and statistics), but the authors fail to take the next step and show it's value and usefulness for management and conservation (despite mentioning it in the Abstract).

Response: Thank you very much for your advice. In order to explain why this study was conducted and how the results of this study can be used for management purposes, we have added a conclusion section later in the discussion. "The results showed that the altitude had a significant effect on the above-belowground biomass distribution. Altitude not only directly affects forest biomass allocation, but also indirectly affects BGBP through other biological and abiotic factors, and the direct effect is greater than the indirect effect. Compared with other influencing factors, climate factors determine the elevation gradient pattern of forest above-belowground biomass allocation distribution. The results of our study will enrich the understanding of the effect of elevation gradient change on forest growth and provide references for the formulation of forest management strategies at different altitudes in the future" (Line392-400).

Round 2

Reviewer 1 Report

Comments and Suggestions for Authors

The authors answered to all questions rised during the first review round therefore, in my opinion, the manuscript is now suitable for publication.

Author Response

Thanks for your hard work and recognition.

Reviewer 2 Report

Comments and Suggestions for Authors

I appreciate many of your responses, but I still have two issues:

First, changing the text to read 1m from the original 50cm for measuring root biomass is good.  However, it is not apparent that any of the calculations or analyses reflect this methodological change.  For each tree, doubling the depth will increase the root biomass used in the calculations, which will increase BGBP.  Yet the graphs look identical from the original.  Also, since many of your data points come from the literature, I question whether you had access to the original raw data to even be able to make this methodological change and do it appropriately.

Regarding multicollinearity, AIC does not address that.  And by adding AIC to the text you have added more issues, particularly with the way you state you are using AIC.  A number of studies in the past have misused AIC, since it is easy to accept the model with lowest AIC as optimal, without assessing the appropriateness of that model.  Which is why current convention with AIC is to consider all models within Delta-2 of the lowest AIC.  Then applying logic and understanding of the biology and ecology of the system being studied to determine which of the models is actually the most appropriate.  And not just blindly accepting the model with lowest AIC as best.  It is not apparent to me that this approach has been used.  There are a number of articles in the literature addressing this issue...

Author Response

Thank you very much for your valuable comments once again. We provide the following responses to the issues you raised:

(1) Your previous review comment suggested a soil depth of 1m, rather than 50cm. My graduate student misunderstood your point, hence the incorrect correction and response. We make the following correction and reply: In actual field work, due to different geographical conditions and variations in soil texture, it is difficult to obtain root biomass at a depth of 1m everywhere. To ensure consistency in our field data collection method and the literature data, we chose a soil depth of 50cm. Of course, we also greatly appreciate your suggestion about the root system differences between coniferous and broadleaf species, which can be a focus for future research.

(2) We used the stepwise regression model approach to address multicollinearity, eliminating redundant variables based on the selected optimal model (the one with the lowest AICc value). This method is well recognized in the scientific community, and we hope it meets your approval.

We apologize again for any inconvenience caused and sincerely hope to receive your affirmation and approval.